# Unraveling Dengue Virus Diversity in Asia: An Epidemiological Study through Genetic Sequences and Phylogenetic Analysis

**DOI:** 10.3390/v16071046

**Published:** 2024-06-28

**Authors:** Juthamas Phadungsombat, Emi E. Nakayama, Tatsuo Shioda

**Affiliations:** Department of Viral Infections, Research Institute for Microbial Diseases, Osaka University, Osaka 565-0871, Japan; juthamas@biken.osaka-u.ac.jp (J.P.); emien@biken.osaka-u.ac.jp (E.E.N.)

**Keywords:** dengue virus, molecular epidemiology, Asia, viral diversity, DENV-1, DENV-2, DENV-3, DENV-4

## Abstract

Dengue virus (DENV) is the causative agent of dengue. Although most infected individuals are asymptomatic or present with only mild symptoms, severe manifestations could potentially devastate human populations in tropical and subtropical regions. In hyperendemic regions such as South Asia and Southeast Asia (SEA), all four DENV serotypes (DENV-1, DENV-2, DENV-3, and DENV-4) have been prevalent for several decades. Each DENV serotype is further divided into multiple genotypes, reflecting the extensive diversity of DENV. Historically, specific DENV genotypes were associated with particular geographical distributions within endemic regions. However, this epidemiological pattern has changed due to urbanization, globalization, and climate change. This review comprehensively traces the historical and recent genetic epidemiology of DENV in Asia from the first time DENV was identified in the 1950s to the present. We analyzed envelope sequences from a database covering 16 endemic countries across three distinct geographic regions in Asia. These countries included Bangladesh, Bhutan, India, Maldives, Nepal, Pakistan, and Sri Lanka from South Asia; Cambodia, Laos, Myanmar, Thailand, and Vietnam from Mainland SEA; and Indonesia, the Philippines, Malaysia, and Singapore from Maritime SEA. Additionally, we describe the phylogenetic relationships among DENV genotypes within each serotype, along with their geographic distribution, to enhance the understanding of DENV dynamics.

## 1. Introduction

The genome of the dengue virus (DENV), an enveloped virus in the Flaviviridae family, consists of a single strand of positive-sense RNA measuring approximately 10.8 kb in length. The single open reading frame encodes a single polyprotein comprising 10 viral proteins: three structural proteins (capsid, pre-membrane/membrane, and envelope) and seven nonstructural proteins (NS1, NS2A, NS2B, NS3, NS4A, NS4B, and NS5). Each of the four DENV serotypes (DENV-1, DENV-2, DENV-3, and DENV-4), which exhibit amino acid differences of approximately 30–35% [1,2], can be further divided into genotypes. Each of these genotypes has no more than 10% diversity in the amino acid levels of the envelope protein [3,4]. DENV is a mosquito-borne virus that is transmitted by *Aedes* mosquitoes, which are the principal vectors. Transmission occurs when an infected mosquito bites a human host. Clinical manifestations of dengue infection range widely, from asymptomatic to severe. In 1997, the World Health Organization (WHO) categorized dengue into three groups based on severity: classic dengue fever (DF), dengue hemorrhagic fever (DHF), and dengue shock syndrome (DSS) [5]. In 2009, the WHO reclassified dengue to facilitate early diagnosis of severe cases. Specifically, dengue was reclassified into dengue without warning signs, dengue with warning signs (abdominal pain, persistent vomiting, fluid accumulation, mucosal bleeding, lethargy, liver enlargement, and increasing hematocrit with decreasing platelets), and severe dengue (severe plasma leakage, severe bleeding, or organ failure) [5,6]. The WHO has also categorized dengue as a neglected tropical disease and has developed strategies for surveillance, prevention, and control to eliminate the disease [7]. Thus, dengue significantly impacts global public health and economies, particularly in tropical and subtropical regions where probable vectors and high population densities exist.

DENV genotypes have historically exhibited distinct geographical distributions in endemic locations [2]. However, the frequency of dengue outbreaks has increased in recent decades, accompanied by changes in epidemiological patterns such as viral extinction, introduction of new genotypes, and replacement of existing genotypes. This review examines the historical and recent epidemiology of DENVs in South Asia as well as Mainland and Maritime Southeast Asia (SEA) from the 1950s to 2023. Additionally, we investigated the phylogenetic relationships among DENV genotypes and their geographic distribution and temporal dynamics in order to enhance our understanding of dengue outbreaks and to develop public health interventions for controlling the spread of the disease.

## 2. Dengue Hyperendemicity in South Asia and Southeast Asia

Dengue-like illness was first documented in South and Southeast Asian regions several centuries ago as a disease that occurred sporadically on a large scale [8]. However, in the 1950s, there was a marked increase in the number of reports of dengue infections. In addition, the terms ‘hyperendemicity’, which refers to the co-circulation of multiple DENV serotypes in a particular area, as well as ‘dengue hemorrhagic fever’ (DHF), which is used to describe severe dengue manifestations, were first used in the 1950s in the post-World War II period [9,10,11,12]. The movement of extensive numbers of troops, an increase in global travel, and ecological disruption promoted the spread of mosquitoes and novel types of dengue across regions and vast distances, all of which facilitated the transmission of dengue [12]. The first recorded DHF epidemic emerged over a wide geographical area, beginning in 1953–1954 in Southeast Asia and then expanding to South Asia. Specifically, the epidemic started with an outbreak in Manila, the Philippines, which mainly affected children [11]. The same clinical symptoms were subsequently observed in Bangkok, Thailand, in 1958; Singapore in 1960; Penang, Malaysia in 1962; and Saigon, Vietnam, and Calcutta, India in 1963 [9,10]. The transmission of dengue in Asia increased throughout the 1980s and 1990s, resulting in high incidence rates and expansion in the extent of epidemic DHF.

Comprehensive surveillance studies undertaken in affected countries in response to the 1950s global pandemic revealed the co-circulation of multiple DENV serotypes [13]. In subsequent decades, hyperendemicity expanded not only in Asia but throughout the world [14]. Pioneering studies early in this period by Hammon et al. and Halsteads et al. identified DENV-2, DENV-3, and DENV-4 during the 1956 epidemic in the Philippines, and DENV-1, DENV-2, and DENV-3 during the 1958 epidemic in Thailand [9,10]. Notably, DENV-3 and DENV-4 were designated as new serotypes at that time. All four serotypes were subsequently isolated during the 1962–1963 epidemic in Thailand [13,15,16], and DENV-1 and DENV-4 were identified in the 1963 epidemic in Cambodia [17]. DENV-2 was identified as being responsible for an outbreak in Vietnam in 1963, and *Aedes aegypti* was found to be the carrier of the virus; all four serotypes were subsequently isolated from mosquitoes after 1967 [18,19]. In the 1976 epidemic in Myanmar, all four serotypes were identified, with DENV-2 and DENV-3 being the most prevalent [20]. In South Asia, the first case of dengue in India in 1945 was identified as being caused by serotype DENV-1 [21]. Other serotypes were isolated over 1956–1967. The first DHF epidemic in India occurred in 1963–1964 and then again in 1967–1968, with DENV-2 and all four types in co-circulation found to be responsible for each epidemic, respectively [20,22,23,24]. In the 1964 epidemic in Bangladesh, DENV-3 was the causative agent. Maritime SEA experienced its first DHF epidemic in 1960 in Singapore and then in 1962 in Penang, Malaysia, with DENV-2 identified as the predominant serotype [25,26]. The first DHF epidemic in Jakarta and Surabaya, Indonesia, occurred in 1968, but serotype information was not available; however, in the 1975–1978 epidemic, Gubler et al. identified four serotypes, with DENV-3 being the predominant serotype [25,27,28,29].

The establishment of DENV endemicity has shown a pattern of recurring outbreaks and fluctuations in the proportions of the co-circulating DENV serotypes [4,30,31]. This pattern is influenced by complex interactions between the virus and host, population immunity, and environmental factors specific to each area. The frequency at which the dominant serotype changes can vary by country and over time. For example, in Thailand, this alternation occurred every 7 to 9 years from the 2000s to 2010s, leading to large epidemic surges every 3 to 5 years [16].

During the 1980s and 1990s, the global transmission of dengue fever experienced a significant surge, leading to an expansion in the geographic distribution of DENV [8]. Early studies on DENV diversity, particularly within serotypes, revealed phylogenetically distinct groups of viruses associated with geographic distributions, referred to as genotypes or subtypes. For example, within DENV-2, genotype II (cosmopolitan) is widespread in South Asia, Maritime SEA, and the Pacific, whereas genotype V is restricted to certain areas in Mainland SEA countries [32]. The classification and distribution of DENV genotypes are discussed in more detail in Section 3, regarding DENV genotypes and classification, and Section 4, regarding the distribution of DENV genotypes in South and Southeast Asia.

In countries that experience dengue outbreaks, the virus is investigated through national surveillance efforts. The data are published and viral sequences are shared in international databases. We collected DENV envelope sequences obtained in 16 endemic countries across three different geographic regions in Asia (Appendix A). These countries included Bangladesh, Bhutan, India, Maldives, Nepal, Pakistan, and Sri Lanka from South Asia; Cambodia, Laos, Myanmar, Thailand, and Vietnam from Mainland SEA; and Indonesia, the Philippines, Malaysia, and Singapore from Maritime SEA. Figure 1 shows the distribution of DENV in these countries over time. Data were more comprehensive for countries such as Singapore, Vietnam, Thailand, Malaysia, and Indonesia and were sparse for Nepal, Pakistan, and the Maldives. Nevertheless, fluctuations in the prevalence of serotypes were apparent in all of these countries.

## 3. DENV Genotypes and Classification

DENVs are classified into four distinct serotypes (DENV-1, DENV-2, DENV-3, and DENV-4). Initially, intra-serotype diversity was categorized into groups referred to as subtypes, which are now commonly referred to as genotypes. This classification, proposed by Rico-Hesse in 1990 [33], grouped the genotypes of DENV-1 and DENV-2 using criteria that assigned DENVs exhibiting no more than 6% sequence divergence within the small regions of the E and NS1 proteins (240 bp) into the same group [33]. Subsequently, classification of DENV genotypes has been based only on the E gene region, which is 1485 bp for DENV-1, DENV-2, and DENV-4 and 1479 bp for DENV-3 [34,35,36]. The development of whole-genome sequencing, which has become commonplace, has facilitated retrospective studies on the DENV stocks collected in previous decades. This practice has significantly enhanced our understanding of DENV evolution and diversity [37,38]. Analyses of both the E gene and the complete genome have provided concordant results [38]. However, the application of whole-genome sequencing has inherent limitations, such as the absence of sequences for certain genotypes and the occurrence of recombination events that may lead to erroneous genotyping. Although DENV has four serotypes and coinfection has been documented, recombination events are relatively rare, especially inter-serotype recombination. However, there is evidence of intra-serotype recombination [1].

The classification of DENV genotypes using phylogenetic methods generally involves several key steps: preparing a nucleotide alignment of DENV sequences with reference strains representing each DENV genotype, constructing a phylogenetic tree, and then distinguishing genotypes by clustering the viruses. An optional step in this process involves estimating sequence divergence by calculating the mean pairwise distance among sequences. A critical component of this method is the selection of the reference strains, which might vary across different studies. In the present study, we randomly sampled sequences from NCBI Virus (https://www.ncbi.nlm.nih.gov/labs/virus/vssi/#/ (accessed on 19 January 2024)), selecting 20 sequences per country per year. Sequence alignment was conducted in AliView v1.28 and executed using MAFFT v.7. To analyze phylogenetic relationships, the maximum likelihood tree of each DENV was constructed using IQ-TREE multicore v2.2.2.6. Furthermore, to assign DENV genotypes and obtain accurate and consistent results, we employed the Genome Detective dengue typing tool [39] and reviewed the literature pertaining to dengue diversity and genotype classification (Figure 2, Figure 3, Figure 4, Figure 5, Figure 6, Figure 7, Figure 8 and Figure 9). The Genome Detective dengue typing tool uses Blast and phylogenetic methods to identify serotypes, genotypes, and lineages from nucleotide sequences. The obtained findings corroborated the classifications of Rico-Hesse et al. for DENV-1 and DENV-2 [33], Twiddy et al. for DENV-2 [32], Chungue et al. for DENV-3 [40], and Klungthong et al. for DENV-4 [41]. Using the topology of our phylogenetic tree, the obtained genotype sequences were assigned to specific clades and lineages (Figure 3, Figure 5, Figure 7, Figure 9 and Appendix A).

## 4. Distribution of DENV Genotypes in South and Southeast Asia

### 4.1. DENV-1 Genotype

The initial classification of the DENV-1 serotype by Rico-Hesse [33] revealed five genotypes. Goncalvez et al. [42] redefined this classification based on 44 complete envelope gene sequences, and subsequent studies have analyzed thousands of sequences from around the world. For example, Chen and Vasilakis [34] examined 1812 envelope sequences published before 2008, while Villabona-Arenas and Zanotto [43] elucidated 1583 envelope sequences (data up to January 2009) from 45 different locations. More recently, Li et al. [44] elucidated the E gene sequences of 5003 samples from 78 epidemic locations. The findings of these studies collectively categorized DENV-1 into five genotypes, which was consistent with genome-based assignment. Unfortunately, whole-genome coverage for genotype II was lacking [38]. The most recent genotype, VI, was proposed in 2014 [45]. Although Genome Detective [39] assigned DENV-1 into six genotypes, there are some discrepancies in the naming of these genotypes. In summary, DENV-1 has been defined into six genotypes, as follows:(1)Genotype I (DENV-1I):

The oldest strains, Hawaii [46] and Mochizuki [47], circulated in the 1940s. The majority of DENV-1I strains are distributed in Mainland SEA and China, but also in Australia, Saudi Arabia, and Somalia.

(2)Genotype II (DENV-1II):

DENV-1II was previously distributed in Thailand in the 1950s and 1960s but is no longer detected.

(3)Genotype III (DENV-1III):

Previous assignment: Malaysian strain detected in 1972, observed in monkeys.

Genome Detective: Mainly distributed in South Asia (India, Pakistan, Sri Lanka, Nepal, and Bangladesh), Singapore, the Middle East (Saudi Arabia and Israel), and Africa (Tanzania, Kenya, and Nigeria).

(4)Genotype IV (DENV-1IV):

Distributed in the Pacific region, including French Polynesia, Fiji, East Timor, Papua New Guinea, New Caledonia, and Australia; Maritime SEA, including the Philippines and Indonesia; and Indian Ocean islands, such as Reunion, France.

(5)Genotype V (DENV-1V):

Previous assignment: the Americas, the Caribbean, and South Asia (India, Pakistan, Sri Lanka, Nepal, and Bangladesh); Singapore; the Middle East (Saudi Arabia and Israel), and Africa (Tanzania, Kenya, and Nigeria).

Genome Detective: Some old Indian strains (1950s–1960s); otherwise, mainly in North America, South America, the Caribbean, and Africa (Congo, Angola, and Cameroon).

(6)Genotype VI (DENV-1VI):

Previous assignment: Bruneian strain collected in 2014.

Genome Detective: Old Indian strains collected in the 1970s and 1980s.

#### DENV-1 Genotypes in the South and Southeast Asian Regions

DENV-1I

DENV-1I, the major genotype in Asia, is endemic to Mainland SEA, especially Thailand and Vietnam, and Maritime SEA and has been reported repeatedly in South Asia (e.g., Sri Lanka). DENV-1I was first identified in Thailand in the 1980s [48], and an older strain (OR389293) collected in the Philippines in 1964 was retrospectively reported. Since then, this genotype has been detected regularly each year as the predominant genotype. Similar patterns of detection have been observed in Laos, Cambodia, Vietnam, and possibly Myanmar, although data are lacking for some periods (Figure 2 and Figure 3). From the 1980s to the 1990s, DENV-1I circulated primarily in Thailand. Phylogenetic analysis revealed at least three Thailand clades; the oldest clade consisted primarily of samples from Thailand (1980–1996), the Philippines (1964), and Myanmar (1999–2002) (Clade 1 in Figure 3 and Appendix A). The second oldest Thailand clade was composed entirely of samples from Thailand between 1990 and 2000 (Clade 2 in Figure 3 and Appendix A). The largest clade of DENV-1I was composed of strains that were descendants of viruses circulating from the 1990s to the present, also from Thailand (Clade 3 in Figure 3 and Appendix A). Evidently, Thailand was the source of DENV-1I from which it spread to neighboring countries during the early 1990s; the DENV-1I genotype from Thailand thus acted as the common ancestor of the latter clades (Figure 3).

**Figure 2 viruses-16-01046-f002:**
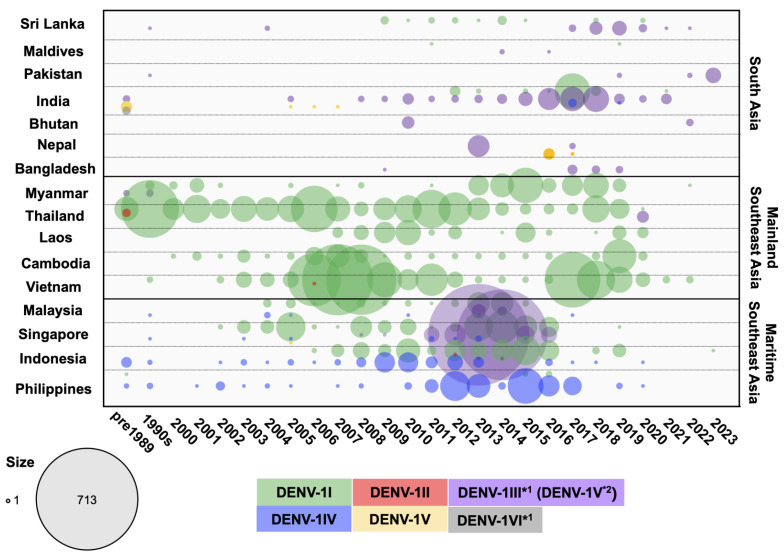
Circulating DENV-1 genotypes in 16 countries across three regions in Asia from the 1950s to 2023. DENV-1 envelope sequences were obtained from GenBank and categorized by genotype and year for each country. The number of DENV cases was aggregated for sequences collected prior to 1989 and for those collected between 1990 and 1999. Differently colored dots represent the genotypes indicated in the legend, with dot size corresponding to the number of sequences. *1 = Genome Detective classification; *2 = Goncalvez [42] and Chen and Vasilakis classification [34].

In the 2000s, DENV-1 was detected frequently and extensively not only in Thailand but also in Vietnam, Cambodia, Laos, and Myanmar. The multiple clusters comprising samples from Vietnam, Cambodia, and Thailand indicate that multiple DENV-1I viruses were introduced into Vietnam and Cambodia in around 1998 and 2000, respectively (Figure 3 and Appendix A). Additionally, the Laos strains (2009–2012) that were included in this cluster were most closely related to Cambodian strains (2006–2008), while those in other clusters were related to Thailand strains that were collected on a similar sampling date (Figure 3 and Appendix A). These findings indicate that these connected regions were characterized by extensive DENV-1I dispersal. Notably, the distribution of the DENV-1I lineage in this region has expanded considerably until the present, and these DENV-1I viruses are relatively closely related.

By the mid-2000s, DENV-1I had newly emerged in several regions, particularly within Maritime SEA, where either DENV-1IV or DENV-1III (previously designated as DENV-1V) had been circulating. For example, Thailand DENV-1I had been introduced to Singapore, Malaysia, and Indonesia by 2002, 2004, and 2006, respectively. The prevalence of DENV-1I increased markedly in Singapore in 2003–2005, 2008–2010, and 2013–2016; in Malaysia in 2013–2014; and in Indonesia in 2007–2010 and 2012–2016. The phylogenetic analysis revealed that the Maritime SEA DENV-1I viruses that were in circulation from 2004 to 2015 grouped into clusters and formed a monophyletic clade that was separate from the previous Mainland SEA lineages, indicating the likelihood of geographical transmission within local regions (Figure 3 and Appendix A). Interestingly, the Laos (2014–2019) and Philippines (2015–2016) clusters belong to this lineage. Furthermore, the more recent Thailand, Myanmar, Laos, and Cambodian strains (2017–2020) and Indonesian strains (2019) formed a cluster within this Maritime SEA lineage (Figure 3 and Appendix A).

**Figure 3 viruses-16-01046-f003:**
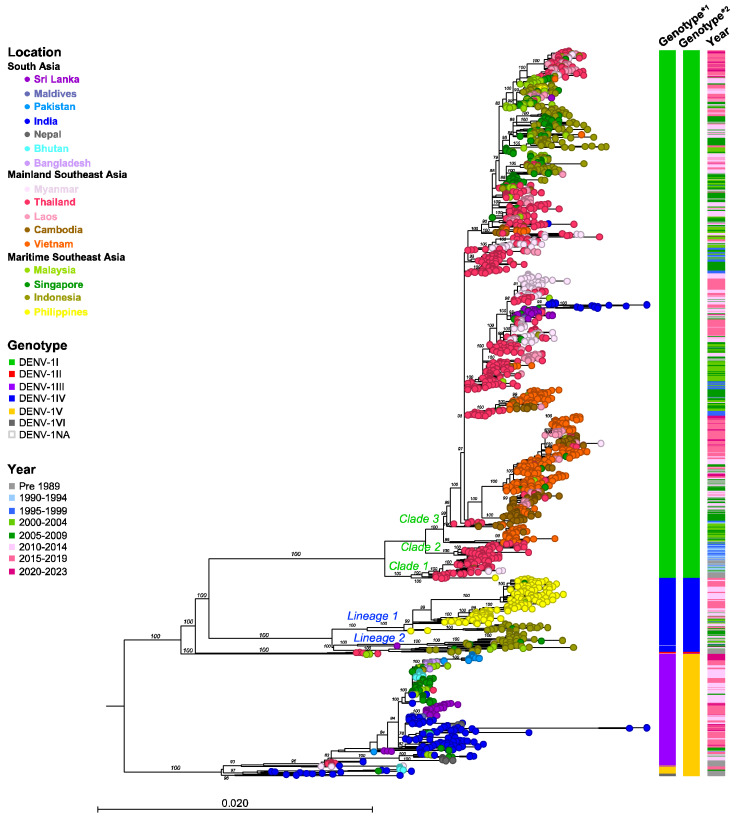
Maximum Likelihood tree of DENV-1 based on the envelope sequence (*n* = 2225). DENV-1 samples were collected from 16 countries across three regions of Asia, with a maximum of 20 sequences collected per year. The nucleotide sequences were aligned using MAFFT v.7 and a phylogenetic tree was constructed in IQ-TREE v.2.2.2.6 using the GTR + F + I + R4 model with 1000 ultrafast bootstrap replicates. Bootstrap values are shown at the tree branches. Each terminal node is colored according to the country of origin. Locations, genotypes, and sampling years are shown in the legend. The scale represents the substitution rate per site. The panel on the right shows the genotypes and range of sampling years corresponding to each taxon. Clades and lineages are indicated adjacent to the key branches, with colors corresponding to each genotype. Genotype*1 = Genome Detective classification; Genotype*2 = Goncalvez [42] and Chen and Vasilakis classification [34].

In the 2010s, a distinct DENV-1I clade comprising sequences from Thailand, Myanmar, Sri Lanka, and India emerged (Figure 3 and Appendix A). These viruses were most likely derived from Thailand DENV-1I, which was collected in the late 2000s. Sequences from Thailand and Myanmar viruses that were collected during 2013–2019 clustered together, suggesting that they might be derived from the same epidemic [49]. On the other hand, DENV-1I was first reported in Sri Lanka in 2009 and was found to be most closely related to strains collected in Singapore in 2009 (Figure 3 and Appendix A). In addition, this virus was associated with DENV-1I from China [50]. DENV-1I was predominant during 2009–2015 and co-circulated with DENV-1III (previously assigned to DENV-1V) in 2018 and 2020 in Sri Lanka (Appendix A). The Sri Lanka DENV-1I subsequently separated into two clusters: a small cluster that contained strains from Sri Lanka (2011–2020) and one strain from Indonesia in 2019, and a large cluster that included strains from Sri Lanka in 2010–2014, Singapore in 2009–2012, and single strains from India in 2012 and Thailand in 2014 (Appendix A). Interestingly, the India cluster (2012–2018) was derived from the Sri Lanka cluster, even though strains from Singapore in 2012 were more closely related (Appendix A). The emergence of DENV-1I in India occurred in the background of the DENV-1III circulation (previously assigned to DENV-1V) (Appendix A). Among the Indian DENV-1I strains that were detected in 2012–2013 and 2015–2018, a large number of DENV-1I cases were reported in the 2017 epidemic in co-circulation with DENV-1III.

DENV-1III and DENV-1V

The phylogenetic trees of DENV-1 in Figure 3 and Appendix A show that the DENV-1V clade consists of Indian strains collected in the 1960s, more recent Indian strains from 2005 to 2007, a strain from Singapore in 2005, and strains from Bhutan in 2016–2017. The cluster of viruses that separated from this clade was assigned to DENV-1III by Genome Detective, but it was still considered to be DENV-1V by the studies described in the previous paragraph [34] (Appendix A). In this clade, the old strains from the 1970s and 1980s that were collected in Myanmar, Thailand, and India were considered to be the ancestral strains. DENV-1III (DENV-1V) is in circulation primarily in all South Asian countries, as well as Singapore and Malaysia. It was less frequently reported in the 1990s and early 2000s. However, since 2005, DENV-1III (DENV-1V) has been steadily increasing in India, peaking in 2016–2018, suggesting that India is the primary area for the source of the virus (Appendix A). Related to the Indian strains collected around the same period, DENV-1III (DENV-1V) strains were also observed in Sri Lanka (2004) and Nepal (2010, 2017, and 2022). Furthermore, the cluster from Sri Lanka, which was associated with a large epidemic in 2019–2020, was found to be related to strains identified in the Maldives (2014 and 2016) and India (2010–2013) (Appendix A).

Meanwhile, in 2009–2011, DENV-1III (DENV-1V) strains from northern India and Bangladesh were introduced into Singapore and Malaysia as well as Bhutan, leading to a large outbreak in 2013–2014 [51]. Within the Singapore/Malaysia clade, clusters from Bangladesh (2017–2018 and 2017–2019) and Pakistan (2019 and 2023) were included (Appendix A). This indicates that the DENV-1III (DENV-1V) strains were reintroduced to the South Asian region.

Furthermore, DENV-1VI included older Indian strains from 1970–1971 and 1985, which formed a cluster related to the DENV-1V genotype.

DENV-1IV

DENV-1IV is the genotype present in the Maritime SEA region, specifically in the Philippines and Indonesia, where the earliest strains were recorded in 1974 and 1984, respectively (Figure 3 and Appendix A). DENV-1IV has circulated continuously within this area up to the present. The phylogenetic tree revealed that DENV-1IV separated into two distinct lineages: the Philippines lineage and the Indonesia lineage (Figure 3 and Appendix A). The DENV-1IV genotype likely originated in the Philippines [43]. The older strains from the Philippines, located at the base of the DENV-1IV Philippines lineage, have given rise to new subclades within the country, comprising strains collected from the 1990s to 2020 (Appendix A). This indicates that the Philippines lineage of the virus has remained geographically restricted and sustained in the area. Remarkably, DENV-1IV has maintained a single genotype in the Philippines over time (Figure 3). In addition, the Philippines DENV-1IV virus was introduced to Indonesia in the early 2000s, where it remained in circulation until 2016 (Appendix A). On the other hand, the Indonesian lineage has been present in Indonesia since the 1980s. It was frequently detected from 2008 to 2010, with fewer cases in the years leading up to 2020. There was also a small spillover into neighboring countries, including Singapore and Malaysia (Appendix A).

### 4.2. DENV-2 Genotypes

The DENV-2 genotype was first classified by Rico-Hesse et al. [33] and Lewis et al. [52], before it was finally defined by Twiddy and Holmes [32] and Chen and Vasilakis [34] based on the analysis of 147 and 1827 envelope sequences, respectively. The genotype names were assigned to each distinct cluster based on its geographical distribution, which were designated as Asian-I, Asian-II, Asian-American, American, Cosmopolitan, and Sylvatic. Similarly, distinct genotypes were proposed by Vu, who analyzed 941 envelope region sequences from human sera collected around the world between 1995 and 2009 [53], and Waman, who analyzed 990 complete genomes (as of November 2015) [54]. Currently, six genotypes have been published, with the previous names refined using roman numerals [39]:(1)Genotype I (DENV-2I, American): Originating from older strains from South Asia and the Pacific, found in the Americas.(2)Genotype II (DENV-2II, Cosmopolitan): Found in South Asia, Maritime SEA, and the Pacific.(3)Genotype III (DENV-2III, Asian-American): Circulating in SEA, the Caribbean, and the Americas.(4)Genotype IV (DENV-2IV, Asian-II): Found in the Philippines.(5)Genotype V (DENV-2V, Asian-I): Common in Mainland SEA.(6)Genotype VI (DENV-2VI, Sylvatic): Sylvatic strains collected in specific locations such as Nigeria, Burkina Faso, Senegal, and Malaysia.

#### DENV-2 Genotypes in the South and Southeast Asian Regions

DENV-2I (American)

Almost all DENV-2I (American) strains in South Asia and Southeast Asia were detected in India during the early period from the 1950s to the 1980s, with single strains detected in 2001 and 2007 [55,56] (Figure 4). The Indian DENV-2I sequences were closely related, forming a single clade. Additionally, one Indonesian strain (GQ398257) collected in 1977 was reported in the database and was included in the Indian clade, showing the closest relation to an Indian strain collected in 1971 (Figure 5 and Appendix A). DENV-2I was the first genotype to circulate widely in India prior to 1971, but it was subsequently displaced by DENV-2II (Cosmopolitan) [55].

DENV-2II (Cosmopolitan)

DENV-2II (Cosmopolitan) is the genotype with the most extensive geographical distribution, being endemic and highly prevalent throughout the South Asia and Maritime SEA regions (Figure 4). From the 1950s to the 2020s, DENV-2II (Cosmopolitan) has been reported annually and consistently in India, Malaysia, Singapore, Indonesia, and the Philippines. Additionally, several surges in prevalence have been reported in Bangladesh, Sri Lanka, and Pakistan. As we reported previously, DENV-2II (Cosmopolitan) spread to, and became established in, Mainland SEA in Vietnam [57] and Thailand [58] in the mid-2000s and in Laos and Cambodia in the mid-2010s. However, DENV-2II (Cosmopolitan) did not displace 2V (Asian-I), the genotype native to Mainland SEA. The phylogenetic clade DENV-2II is divided into two lineages with different strains, each with different geographic regions, detected from the early period (Figure 5).

**Figure 4 viruses-16-01046-f004:**
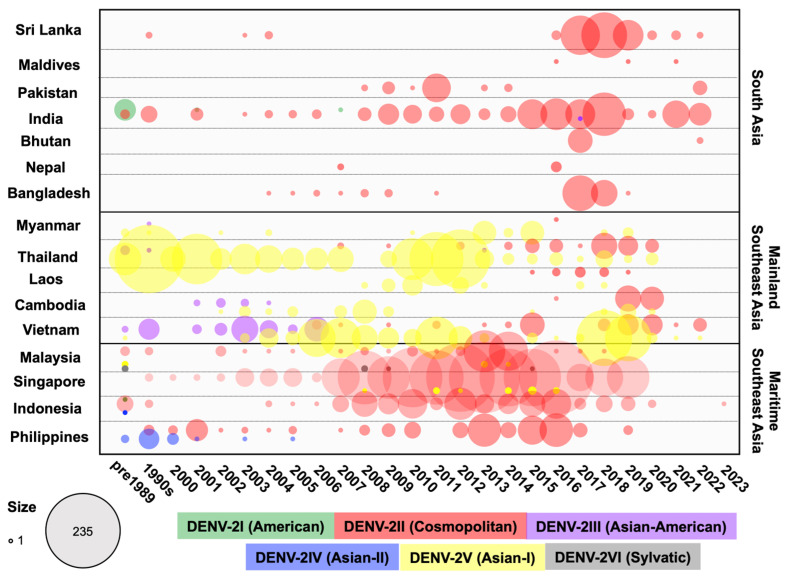
Circulating DENV-2 genotypes in 16 countries across three regions in Asia from the 1950s to 2023. DENV-2 envelope sequences were collected from GenBank and categorized by genotype and year for each country. The number of DENV cases was aggregated for sequences collected prior to 1989 and for those collected between 1990 and 1999. Differently colored dots represent the genotypes indicated in the legend, with dot size corresponding to the number of sequences.

(1)The DENV-2II Indian lineage originated from strains from Malaysia (1969 and 1989) and India (1974, 1983, and 1988), which clustered at the base of this lineage (Lineage 1 in Figure 5 and Appendix A). The emergence of this lineage, particularly in India during the 1980s, led to the complete replacement of the DENV-2I genotype (American) [55]. Following its emergence, the DENV-2II Indian lineage underwent continued diversification, resulting in the formation of three distinct subclades. The first subclade included strains from Sri Lanka (1994 and 1996) and Singapore (1991) ((1) in Figure 5 and Appendix A). The second subclade comprised various clusters predominantly from India (1990–2021), Bhutan (2016), Nepal (2017), and Bangladesh (2017), indicating regional dispersal and localized evolution ((2) in Figure 5 and Appendix A). The third subclade consisted of clusters from India (1991–2022), Bangladesh (2004–2011), Bhutan (2007), Pakistan (2008–2014), and Sri Lanka (2003–2004), suggesting sustained circulation ((3) in Figure 5 and Appendix A). Furthermore, detection of the DENV-2II Indian lineage outside South Asia was noted, with these strains related to those from India. Notable examples include viral clusters from India (2014), Singapore (2014–2015), Malaysia (2014), and Thailand (2013–2015). Another viral cluster included strains from India (2014), Singapore (2012 and 2017), Thailand (2014–2015), Laos (2015–2018), and Vietnam (2014–2015), indicating viral dissemination across regions (Figure 5 and Appendix A).(2)The DENV-2II Asian-Pacific lineage contained Indonesian strains from 1975 to 1976, which served as the common ancestral strains (Lineage 2 in Figure 5 and Appendix A). Throughout the 1990s, the DENV-2II Asian-Pacific lineage virus circulated within Indonesia, Singapore, and Malaysia. By the 2000s, the DENV-2II Asian-Pacific lineage was introduced to the Philippines. The Philippines strains collected from 1998 to 2019 formed a distinct monophyletic clade, indicating a single introduction (Figure 5 and Appendix A). The earliest Philippines strain was related to Malaysian strains collected in 1997, suggesting a possible source–sink relationship between these two regions. Meanwhile, localized Indonesian strains collected from the 2000s to the mid-2010s belonged to distinct clusters, including the Indonesian cluster and the Maritime SEA clusters comprising strains from Singapore and Malaysia collected during a similar timeframe (Figure 5 and Appendix A). Additionally, separate clusters specific to the Singapore or Malaysian strains were detected, indicating endemic circulation within this region (Appendix A). The DENV-2II Asian-Pacific lineage virus was reported annually in Indonesia, Singapore, Malaysia, and the Philippines over this period. Periodic outbreaks occurring at intervals suggest the existence of reservoirs or hotspots within these areas [59,60,61].

**Figure 5 viruses-16-01046-f005:**
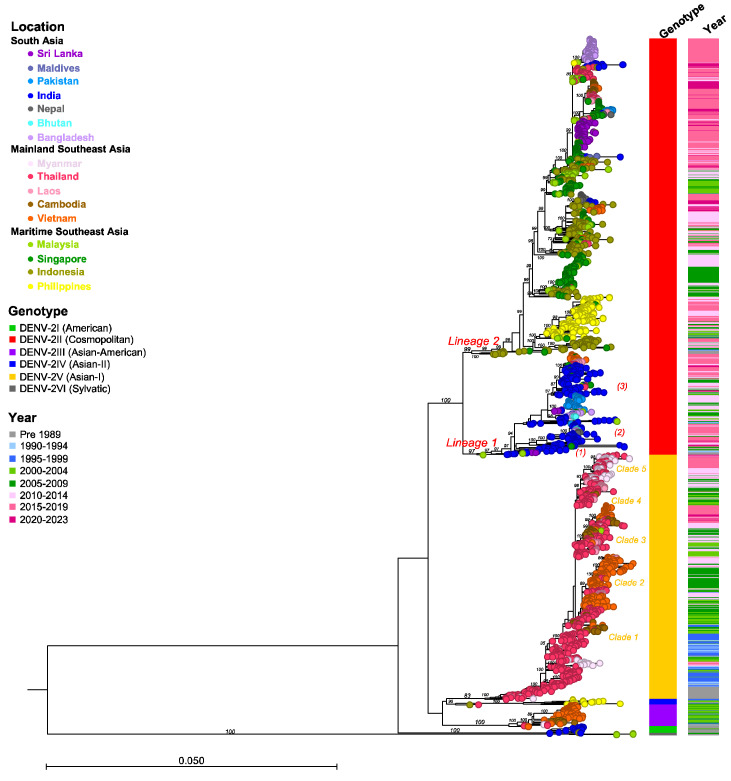
Maximum Likelihood tree of DENV-2 based on envelope sequences (*n* = 2375). DENV-2 samples were collected from 16 countries across three regions of Asia, with a maximum number of 20 sequences per year. The nucleotide sequences were aligned using MAFFT v.7 and a phylogenetic tree was constructed in IQ-TREE v.2.2.2.6 using the GTR + F + I + R6 model with 1000 ultrafast bootstrap replicates. Bootstrap values are shown at the tree branches. Each terminal node is colored according to the country of origin. Locations, genotypes, and sampling years are shown in the legends. The scale represents the substitution rate per site. The panel on the right shows the genotypes and range of sampling years corresponding to each taxon. Lineages are indicated adjacent to the key branches, and clades and subclades in parentheses are indicated adjacent to the key clusters, with colors corresponding to each genotype.

During the late 2010s, phylogenetic analysis revealed extensive spreading and diversification of the DENV-2II Asian-Pacific lineage across Mainland SEA and South Asia resulting from multiple introductions (Appendix A). Viruses isolated from different countries during this period formed various co-circulating clades: the Nepal 2017 clade and the India 2022 clade; the Vietnam 2019–2020 clade (Appendix A); the Maldives 2016, 2019, and 2021 clades, the India 2018 clade, and the Sri Lanka 2016–2022 clade (Appendix A); and the India 2021 clade, the Pakistan 2022 clade, and the Singapore 2018–2019 clade (Appendix A). Additionally, the new emerging Mainland SEA clade consisted of viruses collected in Thailand from 2016 to 2019, Laos from 2017 to 2018, Cambodia from 2019 to 2020, and Vietnam from 2019 to 2022. Notably, this clade also included the Thailand 2015–2020 clades, the India 2018–2022 clade, and the Bangladesh 2017–2019 clade (Appendix A). Interestingly, each of these clades showed close genetic associations with viruses isolated from Indonesia, Singapore, and Malaysia during the 2010s. These findings suggest that these countries may have been possible source regions for viral dissemination and transmission.

The emergence of the DENV-2II Asian-Pacific lineage in naïve regions resulted in a surge in case numbers. This was observed, for example, during the 1998 epidemic in the Philippines, where the DENV-2II Asian-Pacific lineage co-circulated with DENV-2IV (Asian-II), the localized genotype, before displacement occurred in 2001–2002 [62,63]. In the case of epidemics in Bangladesh, Sri Lanka, and Nepal in 2017, the DENV-2II Asian-Pacific lineage emerged against the background of the Indian lineage [64,65,66].

DENV-2III (Asian-American)

DENV-2III (Asian-American) was first detected in the 1980s in Southeast Asia, particularly in Thailand and Vietnam, before spreading to the Americas in the early 1980s [67,68] (Figure 4). This genotype successfully displaced the previously existing American genotype and has persisted in the Americas until the present day. In contrast, the circulation of DENV-2III (Asian-American) in Asia was reported until the mid-2000s, after which the genotype was replaced by DENV-2IV (Asian-I) [53]. Phylogenetic analysis revealed that viruses isolated in Thailand, Vietnam, and Myanmar during the 1980s and 1990s, as well as those from Vietnam between 1997 and 2005 and Cambodia between 2001 and 2004, clustered together (Figure 5 and Appendix A).

DENV-2IV (Asian-II)

DENV-2IV (Asian-II) had limited circulation in Indonesia in 1975 and in the Philippines from 1983 to 2005 (Figure 4). Within the Philippines, DENV-2IV (Asian-II) was the first DENV-2 genotype detected and was reported frequently in 1983, 1988, and 1994–2001, as well as in travelers visiting the Philippines in 2003 and 2005 [69,70] (Figure 5 and Appendix A). Additionally, it was associated with outbreaks and epidemics [69,71]. However, after the emergence of DENV-2II (Cosmopolitan) in 1998, DENV-2IV (Asian-II) declined and eventually disappeared from 2006 onward [62].

DENV-2V (Asian-I)

DENV-2V (Asian-I) originated in Mainland SEA and has remained endemic and restricted to this region up to the present (Figure 4). In the early period, DENV-2V (Asian-I) strains were consistently detected in Thailand annually between 1964 and 1988, with three strains in Myanmar obtained in 1976 and 1985 and one strain obtained in Vietnam in 1988. The first Thailand DENV-2V (Asian-I) strain was recovered from a pediatric patient with DHF during the 1964 epidemic in Bangkok [72]. Moreover, DENV-2 was the most prevalent serotype in 1973–1986 and 1988–1989 in Thailand [13,48]. From the mid-1990s to the mid-2000s, DENV-2V (Asian-I) was primarily detected within Thailand, with these strains grouping into clusters that differed by collection time. Notably, DENV-2V (Asian-I) was detected frequently in Thailand, with periodic epidemics and outbreaks characterized by marked increases in case numbers in the early 2000s [31]. DENV-2V (Asian-I) was the sole genotype circulating in Thailand from 2000 to 2007.

From the mid-1990s, at least four to five major clades were established, with Thailand strains from the 1990s being the common ancestor of each clade (Figure 5 and Appendix A). This suggests that Thailand likely acted as the source of DENV-2V (Asian-I) from where it subsequently spread to Myanmar, Laos, Cambodia, and Vietnam.

Clade 1 (Cambodia clade): In Cambodia, DENV-2 was the dominant strain from 2002 to 2005 with multiple genotypes. The local DENV-2III (Asian-American) and the newly emerging DENV-2V (Asian-I) [53,73] Cambodia strains from 2002 to 2008 and Vietnam strains (2004 and 2010) were grouped with the Thailand strains from 2000 to 2004, indicating a shared origin with the Thailand strains from the 1990s (Appendix A).

Clade 2 (Vietnam clade): This clade contained a large number of Vietnam strains circulating from 2003, when DENV-2V (Asian-I) first emerged in the country [53], to 2017, appearing in Thailand from 2003 to 2013 and in Laos from 2012 (Appendix A).

Clade 3: This clade consisted of several subclades, including strains from Laos and Thailand that were collected during 2007–2011 and from Cambodia and Thailand during 2008–2014. Moreover, recent strains from 2017 to 2022 that were associated with reemergences in Thailand and Cambodia in 2018–2020, Laos in 2017, and Vietnam in 2017–2022 were grouped together to form a distinct clade (Appendix A).

Clade 4: This clade included strains from Thailand (2005–2014), Laos (2008–2010), and Myanmar (2013–2015) (Appendix A).

Clade 5: This clade represented a recently separated cluster of strains from Thailand and Myanmar that circulated in 2015–2020 (Appendix A).

### 4.3. DENV-3 Genotypes

Based on an analysis of 23 prM/M and E gene sequences of DENV-3 strains collected in four different geographic regions from 1956 to 1992 [35] and 88 complete E genes collected from 1956 to 2005, Weaver [36] characterized four distinct DENV-3 genotypes (I–IV). An additional genotype, V, was subsequently distinguished by Wittke [74], which included viruses that were previously classified as Genotype I (i.e., Philippines H87 1956, China 1980, and Malaysia 1981). These genotypes are consistent with those defined by Chen [34], Araujo [75], and Waman [76], who analyzed a large nucleotide dataset consisting of 1208 envelope sequences and 860 whole genomes. However, the entire genome of DENV-3IV has not yet been sequenced. Finally, the DENV-3 genotypes and their epidemiological characteristics are as follows:(1)Genotype I (DENV-3I): Identified in Maritime SEA regions such as Malaysia, the Philippines, and Indonesia, as well as the Asia-Pacific islands.(2)Genotype II (DENV-3II): Initially identified in Thailand in the 1960s and in India and Sri Lanka in the 1980s. These viruses have since been maintained in Mainland SEA and South Asia, particularly in Bangladesh.(3)Genotype III (DENV-3III): The oldest strains were identified in India and Sri Lanka in the 1960s and 1980s, respectively [77]. DENV-3III has continued to be detected in this region and in Maritime SEA (Malaysia and Singapore). It has also spread across the Americas, including Brazil, Venezuela, Nicaragua, Cuba, Puerto Rico, and Peru, and has recently been detected in Africa.(4)Genotype IV (DENV-3IV): Detected inadequately, with only a few sequences reported from Tahiti in 1965 and Puerto Rico in the 1960s and 1970s.(5)Genotype V (DENV-3V): Contains the Philippines strain H87 (1956) and strains from China (1980) and Brazil (2002).

#### DENV-3 Genotypes in the South and Southeast Asian Regions

DENV-3I

DENV-3I is endemic to Maritime SEA, particularly Indonesia and the Philippines (Figure 6 and Figure 7). The basal clade of DENV-3I includes Indonesian strains associated with outbreaks in the 1970s in Java [27,78] and in the 1980s in Jakarta [79], as well as Malaysian strains collected in 1981 [35] and Philippines strains collected in 1983 (Figure 7 and Appendix A). Earlier DENV-3I viruses subsequently diverged into two major clades:(1)Clade 1: Almost all of the Philippines DENV-3I strains that were detected between 1983 and 2020 grouped into a monophyletic clade. This clade also contains a Malaysian strain observed in 2009 and clusters of Singapore strains from 2008 to 2009 and 2013 to 2014. DENV-3I in the Philippines was reported to be the only genotype in sporadic annual outbreaks, but cases increased significantly in 2005–2006, 2009 [80,81,82], 2012, and 2015–2017 [83].(2)Clade 2: A large DENV-3I clade consisting of several countries, particularly Indonesia, the source country Malaysia, Singapore, Sri Lanka, Bangladesh, Myanmar, and Thailand, where the virus was recently introduced in the mid-2000s. A single Malaysian strain (L11429.1) that was first documented in 1974 was observed at the base of this clade (Figure 7 and Appendix A). In the 1980s, the virus was primarily detected in Indonesia, where it became the predominant genotype. The 1998 epidemic in Sumatra, Indonesia, was caused by DENV-3I [59]. The Indonesian strain from 1998 (AB189128) was related to those collected in the earlier period of Indonesia. Two Malaysian strains from 1990 and 1991 clustered within the Indonesian clade from 1988 and 1991 (Appendix A). Furthermore, two Malaysian strains from 1997 were collected from cities in Sarawak, bordering Indonesia [84]. Another country-wide epidemic occurred in Indonesia in 2004, and although it was caused by several serotypes, DENV-3 was particularly widespread in Jakarta and Bandung [85,86,87]. DENV-3I was widely sampled throughout the period from 2007 to 2014 [88,89,90,91], with a spike during an outbreak in 2015–2016 [92,93,94]. In 2019, a notable outbreak occurred in Manado, North Sulawesi, predominantly involving DENV-3 [95], with DENV-2 observed at other locations [96]. In Malaysia and Singapore, DENV-3I was constantly reported in small numbers from the 2000s to the mid-2010s [97,98,99,100], and these strains clustered together with Indonesian strains from the same period (Appendix A). From 2015 onwards, DENV-3I was first observed in new regions: Mainland SEA, including Thailand [101,102] and Myanmar [103,104], and South Asia, including Bangladesh [66,105] and Sri Lanka [106]. The phylogenetic tree shows that DENV-3I viruses collected in Thailand, Myanmar, Bangladesh, and Sri Lanka between 2015 and 2022 were phylogenetically related to earlier strains from Maritime SEA (Figure 7 and Appendix A), suggesting that these regions were the origin of the newly introduced virus.
DENV-3II


The DENV-3 genotype II was a major genotype in all Mainland SEA countries and in Bangladesh, dominating until around the 2000s before experiencing a sustained decline after the mid-2010s (Figure 6). The earliest DENV-3II sequence was from Thailand in 1962 [35], and it was frequently detected in the 1970s and 1980s, when it was associated with outbreaks [16,48]. Additionally, DENV-3II was present in Myanmar in 1973 and 1976, Sri Lanka in 1983, India in 1984, and Malaysia in 1987–1989. Phylogenetic analysis showed that these sequences from the 1970s and 1980s clustered at the base of the DENV-3I clade (Figure 7 and Appendix A). In the 1990s, DENV-3II was predominant in Thailand, causing severe epidemics with vast numbers of cases, particularly in 1997 and 1998 [16,48,107], leading to the virus spreading to other countries. In the phylogenetic analysis, DENV-3II was split into two major clades.

Clade 1: This large clade consists of clusters of viruses collected from 1986 to 2013, mainly in Thailand, Cambodia, and Vietnam. Thailand strains from the 1980s were the common ancestors (Appendix A). From the 1990s to the 2010s, in addition to Thailand, which experienced repeated DENV-3 endemic/epidemic cycles [4,48], DENV-3II was first detected in 1999 (KF955461) in Cambodia and became prevalent there in early 2000 (Appendix A). The number of cases subsequently increased from 2005 and caused a large epidemic, likely encompassing the entire country in 2007 [73,108,109]. Early Vietnam strains from 1996 to 1998 were related to a Thailand strain from 1996, while the majority of DENV-3II strains circulating in the 2000s were related to the Cambodian strain in 1999, indicating that the DENV-3II invasion likely occurred via Thailand and Cambodia (Appendix A). Furthermore, Malaysian strains identified during outbreaks in 1991–1995 and 2002 [100] clustered with Thailand strains collected during the same period (Appendix A).

Clade 2: This clade, comprising viruses from Thailand, Myanmar, Bangladesh, and Laos and circulating from 1994 to 2014, formed distinct clusters; for example, a cluster of Bangladesh DENV-3II viruses that circulated from 2000 to 2009 (Appendix A). The outbreak began in Bangladesh in 2000 and caused more than 5551 hospitalized cases and 93 fatal cases [110,111]. The older strains that were most closely related to this cluster were from Thailand in 1997 and Myanmar in 1998. In addition, Singapore (2008–2009) and Philippines (2010) viruses were observed in this cluster (Appendix A). Another cluster consisted of DENV-3II strains from Thailand (1994, 1998, and 2005–2013), Myanmar (2005–2009), and Laos (2010–2014). In the case of Myanmar, the 1998 epidemic was caused by multiple genotypes [112]. However, during the 2004–2007 epidemic, DENV-3II was the predominant serotype [113]. These Myanmar DENV-3II strains exhibited close genetic relatedness to Thailand DENV-3II strains from 1998 and 2005–2011 (Appendix A). Laos experienced a significant increase in DENV-3II from 2010 onward, peaking in 2012. The epidemic ended in 2013, with the co-circulation of multiple DENV-3 genotypes [114,115].

Notably, within the DENV-3II endemic region, DENV-3II has not been present in Cambodia after 2008 [73], Myanmar after 2010, and Bangladesh after 2010, as we reported previously [66]. DENV-3II was most recently detected in Vietnam in 2013 [116] and in Thailand and Laos in 2014 [4]. In countries where DENV-3II is not endemic, such as Indonesia, DENV-3II has been extinct since the 1980s, except for one Indonesian strain reported in a travel-associated patient returning to Taiwan in 1998 [117]. The last Malaysian strain was detected in 2002 [99,118], and a single strain was detected in Singapore in 2013 [119].

**Figure 6 viruses-16-01046-f006:**
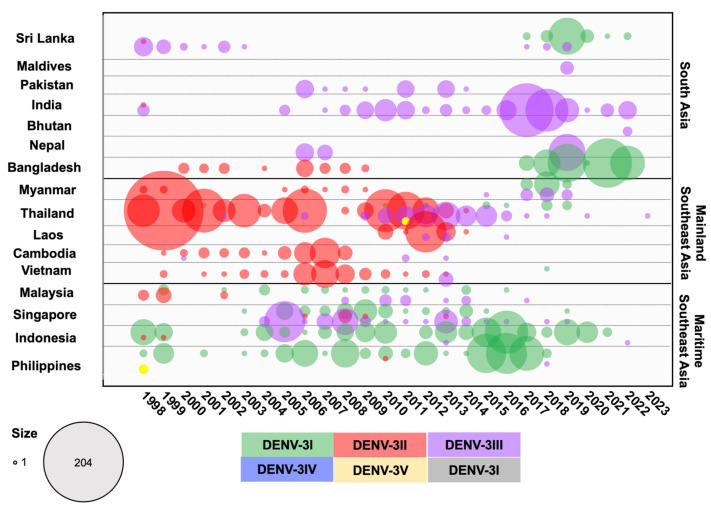
Circulation of DENV-3 genotypes in 16 countries across three regions in Asia from the 1950s to 2023. DENV-3 envelope sequences were collected from GenBank and categorized by genotype and year for each country. The numbers of DENV cases were aggregated for sequences collected prior to 1989 and for those collected between 1990 and 1999. Differently colored dots represent the genotypes indicated in the legend, with the dot size corresponding to the number of sequences.

**Figure 7 viruses-16-01046-f007:**
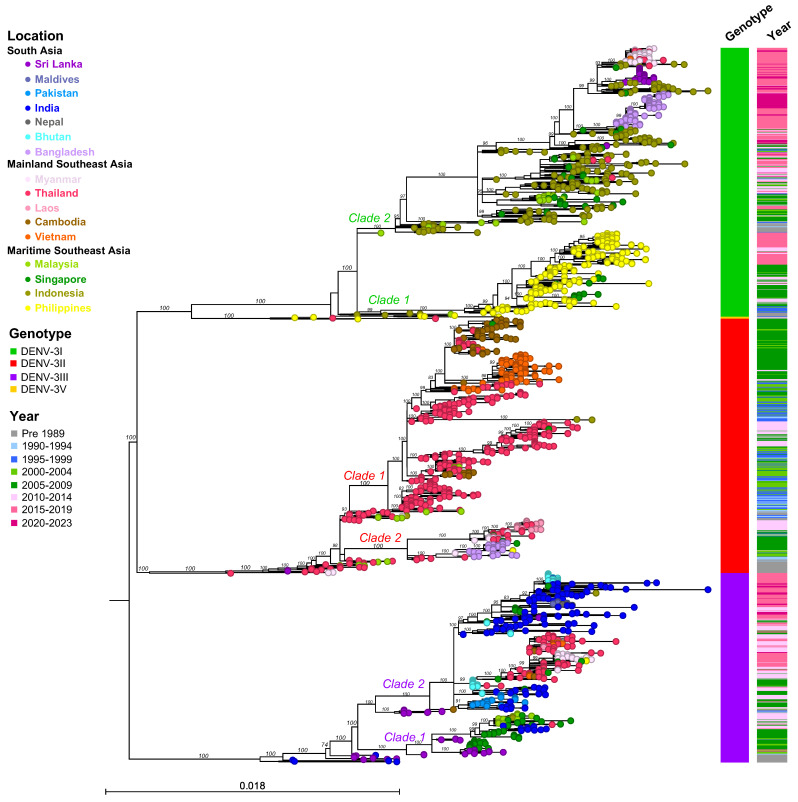
Maximum Likelihood tree of DENV-3 based on envelope sequences (*n* = 1651). DENV-3 samples were collected from 16 countries across three regions of Asia, with a maximum number of 20 sequences per year included in the analysis. The nucleotide sequences were aligned using MAFFT v.7 and a phylogenetic tree was constructed in IQ-TREE v.2.2.2.6 using the GTR + F + I + R6 model with 1000 ultrafast bootstrap replicates. Bootstrap values are shown at the tree branches. Each terminal node is colored according to the country of origin. Locations, genotypes, and sampling years are shown in the legend. The scale represents substitutions rate per site. The panel on the right shows genotypes and the range of sampling years corresponding to each taxon. Clades are indicated adjacent to the key branches, with colors corresponding to genotypes.

DENV-3III

DENV-3III originated on the Indian subcontinent and became endemic in India, Sri Lanka, Pakistan, and Bhutan, while also circulating in Singapore and Malaysia. More recently, DENV-3III has spread to Mainland SEA (Figure 6 and Figure 7). The earliest recorded strain of DENV-3III was isolated in Vellore, Tamil Nadu, India, in 1966, with additional isolates reported between 1966 and 1970 [21,23,24,120]. From the 1980s to the 1990s, Sri Lanka was the center of DENV-3III activity [77,121], and viruses collected in Sri Lanka at this time were the common ancestors of descendant clades that subsequently emerged in India, Pakistan, Bhutan, and Singapore in the 2000s. DENV-3III epidemics first emerged on the Indian subcontinent from 2002 to 2003, initially in Sri Lanka [122] or India. Specifically, India reported more than 10,000 cases of DENV in 2003, as well as in 2005–2006 and 2008–2009 [123,124,125,126]. Additionally, occurrences were noted in Bhutan from 2005 to 2006 [127] and in Pakistan during 2005–2007 [128,129,130]. DENV-3III persisted in the region, with occasional reemergence in subsequent decades [131,132,133]. The spread of DENV-3III expanded to new regions from the 2000s to the 2010s, moving from Malaysia, Thailand, and Laos into Cambodia, Vietnam, and Myanmar.

The circulation of DENV-3III between the 1980s and 2023 can be divided into two major phylogenetic clades.

Clade 1: Evidence of this clade implies that the introduction of Sri Lankan strains to Singapore occurred in the 2000s, followed by India and Malaysia. Notably, DENV-3III was identified as the major DENV-3 genotype of the 2005 epidemic in Singapore [98] and was related to the strain responsible for the epidemic in Sri Lanka that was collected in 2004 [122]. Several subclusters were observed, including the Sri Lanka and Singapore 2004–2019 cluster, the Singapore and India 2007–2011 cluster, and the Malaysia and Singapore 2008–2014 cluster (Appendix A).

Clade 2: This large DENV-3III clade consisted of viruses from South Asia, Mainland SEA, and Maritime SEA regions, which clustered into several distinct subclades. Sri Lankan strains from 1989 to 1991 and 1997 were the ancestral strains of this clade. Notably, Indian strains not only formed individual clades observed in several periods (2005, 2009–2012, 2010–2017, and 2017–2022) but were also included in other clades, such as the Pakistan 2006–2014 clade, the Maldives and Nepal 2016–2022 clade, the Singapore 2012–2013 clade, and the Bhutan 2019 clade (Appendix A). These findings suggest that DENV-3III has been in circulation and predominant in several parts of India and within the South Asian region [120,134,135].

Interestingly, DENV-3III emerged in Mainland SEA around 2009 and has been circulating until the present. In the Bhutan cluster (2006–2007) associated with the outbreak in 2005–2006, DENV-3III was the dominant strain [127], with a Thailand strain in 2002 (MW946958.1) being the common ancestral strain of this emergent clade. DENV-3III was first detected in Thailand in 2002 and been localized there ever since; the highest number of cases attributed to this strain was observed in 2013. DENV-3III co-circulated with DENV-3II, the localized strain, during 2009–2014 and with DENV-3I during 2016–2019. Subsequently, DENV-3III spread to Cambodia in 2011, Laos in 2012, Vietnam in 2013, and Myanmar in 2015 and 2017–2019 [115]. In addition, Singapore strains also formed part of this clade (Appendix A). A single branch of a Singapore strain from 2013 was observed to be related to Thailand strains. From 2015, DENV-3III was encountered less frequently in Southeast Asia, except for Myanmar. However, small clusters of strains from Singapore, the Philippines, and Malaysia (2015–2018) were related to strains from Myanmar (2017–2019) and Thailand (2020 and 2023) (Appendix A).

DENV-3V

DENV-3V was initially identified in the Philippines in 1956 (strain H87) [10,136]. After being absent from Asia for a period, DENV-3V reappeared in 2011 when two strains were detected in A. aegypti collected in urban centers in Thailand, marking the first time that this genotype was detected in this country [137] (Appendix A).

### 4.4. DENV-4 Genotypes

Lanciotti et al. [138] initially described two genotypes based on the envelope sequences of DENV-4 strains from 19 locations between 1956 and 1986. Subsequently, more DENV-4 genotypes were discovered, including Sylvatic strains from Malaysia [41,139]. The classification of DENV-4, as defined by Klungthong et al. [41], was eventually organized into four genotypes, which was in agreement with the larger envelope datasets and whole genomes [34,140]. This classification was as follows:(1)Genotype I (DENV-4I): Commonly found in Mainland SEA, especially Thailand and Vietnam. It has also been detected in the Philippines, Sri Lanka, India, Pakistan, and China.(2)Genotype II (DENV-4II): This genotype has a broader regional circulation and is typically subdivided into two groups—IIA, which includes strains mainly from Maritime SEA, and IIB, which comprises strains that were distributed in Indonesia in the early period and which are currently prevalent in the Americas and the Caribbean islands [141].(3)Genotype III (DENV-4III): This genotype includes strains collected in Thailand between 1997 and 2001 [41].(4)Genotype IV (DENV4-IV): This genotype is composed of Sylvatic strains collected in Malaysia [139].

#### DENV-4 Genotypes in the South and Southeast Asian Regions

DENV-4I

DENV-4I has been the dominant genotype across all Mainland SEA countries and South Asia (Sri Lanka, India, and Pakistan), particularly in Thailand, Vietnam, and Cambodia over the last several decades (Figure 8). Furthermore, it has also sporadically appeared in Maritime SEA (the Philippines, Singapore, and Indonesia). The phylogenetic tree of DENV-4I generated based on sequences of the envelope region reveals the existence of three lineages with geographical associations (Figure 9 and Appendix A).

First (Clade 1 in Figure 9 and Appendix A), the earliest DENV-4 strain, Philippines 1956, formed a cluster with other Philippines strains from 1964 and 1984 as well as a Sri Lanka strain from 1978 at the base of either the DENV-4I subtree or the DENV-4I Philippines lineage. DENV-4I diverged into a clade consisting of strains from the Philippines during 2003–2013, which was restricted to that country. Among these strains, those strains from 2003 to 2005 were detected in travelers visiting the Philippines, while confirmed local transmission was reported in 2012–2013 [70,116,142].

The second lineage, a South Asian lineage (Clade 2 in Figure 9 and Appendix A), originated from India and circulated from the 1960s to the 1980s [24]. This lineage has continued to circulate within Sri Lanka, India, and Pakistan from the late 2000s to the present. The old Indian strain in this lineage was separated from viruses detected after 1996, indicating viral reintroduction. The reemergence first occurred in India in 1996, then in Pakistan in 2009 and in Sri Lanka in 2003 and 2012–2013 [130,143,144]. Interestingly, the prevalence of Indian strains increased markedly in 2012, 2014–2018, and 2021 [145,146].

The third lineage (Clade 3 in Figure 9 and Appendix A), the Mainland SEA lineage, contains most of the DENV-4I strains that have circulated between the 1990s and the present. The lineage is prevalent in Thailand, Vietnam, Cambodia, Myanmar, and Laos. The phylogenetic trees in Figure 9 and Appendix A show that strains from Myanmar (1976) and Thailand (1977) formed a single branch at the base of the clade, indicating that they are the ancestral strains of this lineage. Remarkably, DENV-4I circulated in Thailand continuously from 1976 to 2020. In the early period, DENV-4I was maintained in Thailand from the 1970s to the 2000s, with annual reports of cases peaking in 1994, 1999, and 2004–2006 [41]. It is considered likely that Thailand acted as the major source of this lineage, and that the virus spread from here into the neighboring countries Laos, Cambodia, and Vietnam, resulting in the development of new subclades through diversification throughout the Mainland SEA region. The Vietnamese sequences, which were observed annually from 2000 to 2019, formed a monophyletic clade, indicating the localization of DENV-4I within the country (Appendix A). Significant increases in case numbers were seen in 2011–2013 and 2016–2019, associated with large epidemics [147]. Furthermore, even in countries with less developed surveillance programs, the presence of the virus was observed for extended periods. For example, the virus was reported in Cambodia in 2000–2013 and 2019–2020; Laos in 2009 and 2013–2019; and Myanmar in 1976, 2006, 2008, 2010, and 2013–2019 (Appendix A).

DENV-4II

DENV-4II has been the dominant genotype in the Maritime SEA region from the 1970s to the present, primarily in Indonesia and the Philippines, while maintaining a smaller and steadier prevalence in Malaysia and Singapore (Figure 8). DENV-4II was subdivided by AbuBakar et al. [141] into DENV-4IIA and DENV-4IIB (Figure 9 and Appendix A). DENV-4IIA is widespread across Maritime SEA countries. From its first detection in Indonesia in the 1980s, DENV-4IIA has been circulating until 2004–2022, indicating that this region acts as a reservoir of the virus [148]. DENV-4IIA was reported in the Philippines in 2004 and 2010–2015, and phylogenetic analysis showed that it formed a distinct clade [149] and suggested that it was introduced from Indonesia (Figure 9 and Appendix A). DENV-4IIA has been detected elsewhere in Maritime SEA, such as in Singapore and Malaysia, where it occurred at the same time in the early 2000s. Notably, Singapore strains recorded from 2008 to 2016 consisted of strains that were related to either the Indonesia or Philippines lineages, whereas DENV-4IIA observed in Malaysia from 2011 to 2014 clustered to an Indonesian lineage (Figure 9 and Appendix A). Transmission in Thailand during 2011–2015 was attributed to viruses in the Philippines lineage (Figure 9 and Appendix A). DENV-4IIB has rarely been observed in Asia, occurring once in Indonesia from the 1970s to the 1980s, while a few cases were reported in Malaysia and Singapore in the 1990s (Figure 9 and Appendix A).

On the other hand, the remaining two genotypes of DENV-4 are uncommon (Figure 9 and Appendix A). DENV-4III, which had only 15 identified strains, was largely confined to Thailand [4,41]. After being detected for the first time in 1996, DENV-4III remained in circulation until 2005, with nine strains reported in Bangkok [150]. DENV-4IV, a Sylvatic strain, was isolated in Malaysia in the 1970s [151].

**Figure 8 viruses-16-01046-f008:**
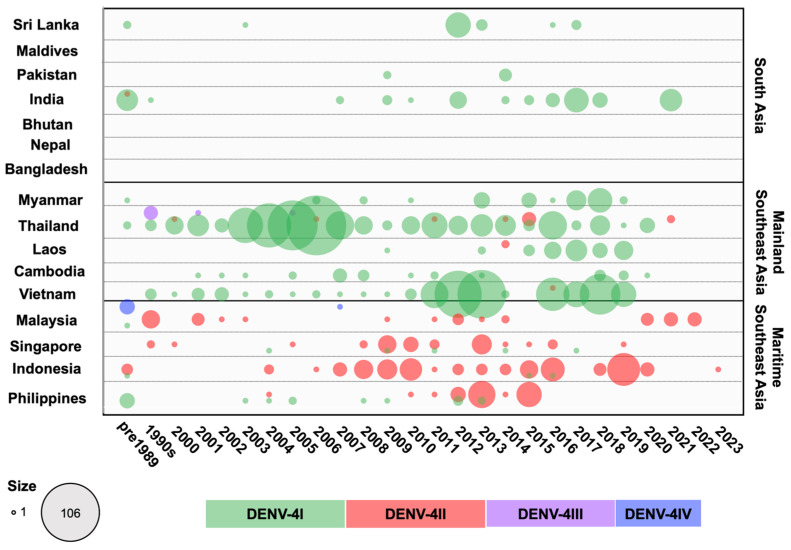
Circulation of DENV-4 genotypes in 16 countries across three regions in Asia from the 1950s to 2023. DENV-4 envelope sequences were collected from GenBank and categorized by genotype and year for each country. The number of DENV cases was aggregated for sequences collected prior to 1989 and for those collected between 1990 and 1999. Differently colored dots represent the genotypes indicated in the legend, with the dot size corresponding to the number of sequences.

**Figure 9 viruses-16-01046-f009:**
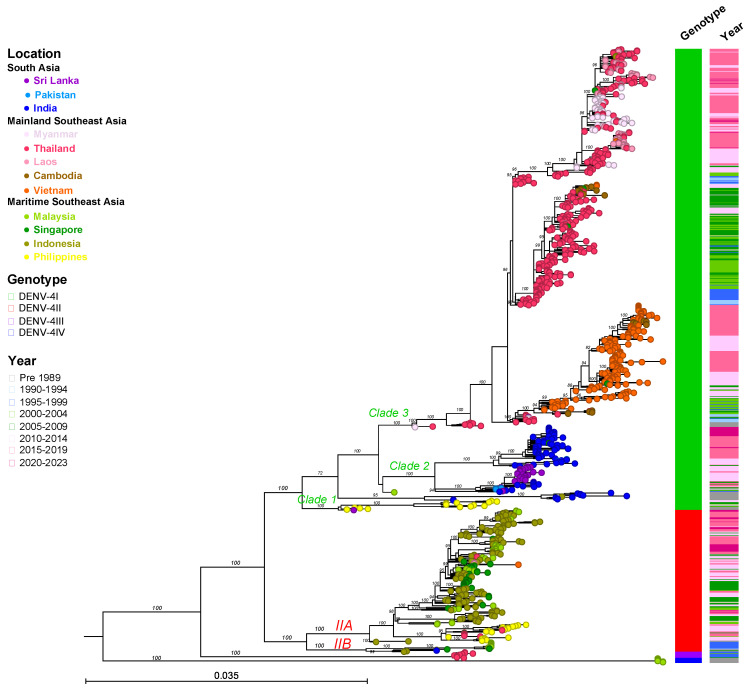
Maximum Likelihood tree of DENV-4 based on envelope sequences (*n* = 1000). DENV-4 samples were collected from 16 countries across three regions of Asia, with a maximum number of 20 sequences per year included in the analysis. The nucleotide sequences were aligned using MAFFT v.7 and a phylogenetic tree was constructed in IQ-TREE v.2.2.2.6 using the GTR + F + I + R6 model with 1000 ultrafast bootstrap replicates. Bootstrap values are shown at the tree branches. Each terminal node is colored according to the country of origin. Locations, genotypes, and sampling years are shown in the legend. The scale bar represents the substitution rate per site. The panel on the right shows genotypes and ranges in sampling years corresponding to each taxon. Clades and groups (IIA and IIB) are indicated adjacent to the key branches, with colors corresponding to each genotype.

## 5. Temporal Dynamics of DENV in South Asia and Southeast Asia

In Asian regions, the co-circulation of multiple DENV types, whether at the serotype or genotype level, presents a complex scenario. Historically, locally circulating strains were often dominated by a single genotype for extended periods, with the dominance of this one genotype characterized by the emergence and disappearance of specific clades. However, this pattern has been disrupted by the introduction of new invasive viral genotypes and lineages, which have become established and remained in circulation. Interestingly, in most countries, these newly emerged types subsequently began displacing or co-circulating with the pre-existing local strains.

In South Asia, the previously predominant DENV types were DENV-1III (DENV-1V), DENV-2II (Cosmopolitan) Indian lineage, and DENV-3III, except in Bangladesh where DENV-3II and DENV-4I were prevalent. However, the introduction of invasive viruses such as DENV-1I, DENV-2II (Cosmopolitan) Asian-Pacific lineage, and DENV-3I has recently been observed in Sri Lanka and Bangladesh (Figure 1, Figure 2, Figure 4, Figure 6 and Figure 8).

In Southeast Asia, particularly in the mainland region, DENV-1I, DENV-2V (Asian-I), DENV-3II, and DENV-4I were predominant, with DENV-2II (Cosmopolitan), DENV-3I, and DENV-3III emerging more recently. In Maritime SEA, Indonesia and the Philippines had historically seen a predominance of DENV-1IV, DENV-2II (Cosmopolitan), DENV-3I, and DENV-4II as the single circulating genotypes at all times. However, DENV-1I spread to Indonesia (Figure 1 and Figure 2). Meanwhile, Malaysia and Singapore previously experienced co-circulation of DENV-1I and DENV-1III (DENV-1V), as well as DENV-3I and DENV-3III (Figure 1, Figure 2, Figure 4, Figure 6 and Figure 8).

The introduction of DENV-1I into Indonesia in the mid-2000s subsequently led to the displacement of DENV-1IV throughout the country [60]. Interestingly, although Indonesia is a maritime nation comprising many islands, the spread of DENV-1I suggests that the geographic barriers imposed by the geographic separation of these islands were easily overcome by human travel. Furthermore, the virus was detected in wild-caught mosquitoes, indicating complete localization and adaptation [152]. Unfortunately, the mechanisms underlying the displacement of one genotype by another have not been well studied. It is likely that DENV-1I has been disseminated across Asia, including India and Sri Lanka, but DENV-1IV remains endemic in the Philippines. Previously, DENV-1I displaced DENV-1II in Thailand in the 1950s and 1960s. The reasons underlying the displacement of DENV-1II from Thailand are unclear, but they are likely related to a lack of virus fitness, or because DENV-1II had been collected from a niche environment [34].

The successful genotype replacement of DENV genotypes was most clearly observed in Thailand, Cambodia, and Vietnam, where DENV-2III (Asian-American) was completely replaced by DENV-2V (Asian-I) in 2006. This shift may be attributed to higher viremia in DENV-2V (Asian-I) patients and a slight difference in virus titer in mosquitoes compared to those infected with DENV-2III (Asian-American), enhancing transmission between humans and mosquitoes [53]. In the Philippines, DENV-2IV (Asian-II) was replaced by DENV-2II (Cosmopolitan). The pre-existing antibodies against DENV-2IV (Asian-II) in the population were found to be less effective in neutralizing the currently circulating DENV-2II (Cosmopolitan). This phenomenon underscores the role of genotype-specific immune responses in driving shifts in the most prevalent virus genotypes [62].

The most recent genotype replacement observed in Thailand involved the replacement of DENV-3II with DENV-3III, a shift likely occurring across the entire Mainland SEA region (Figure 6). The DENV-3III South Asian lineage has been actively circulating in India, Pakistan, and Bhutan. The South Asian DENV lineage has not directly invaded the Mainland SEA region before. In the case of DENV-3, DENV-3II has experienced sustained transmission in the region with recurrent outbreaks, suggesting the development of protective immunity within the community. Significantly, the spread of the DENV-3III South Asian lineage from India to Thailand occurred via Bhutan (Figure 7). Notably, the first transmission of DENV in Bhutan, which occurred in a temperate area, occurred during the summer of 2004 in cities adjacent to India and was significantly affected by climatic factors [153]. Subsequently, DENV-3III was introduced from bordering areas and remained prevalent throughout the country, causing an outbreak during 2005–2006 [127]. The subsequent emergence of DENV-3III in Thailand in the late 2000s exhibited a close genetic relationship with the Bhutan strain (Figure 7 and Appendix A), indicating that the virus might have circulated in the mosquito vector and became disseminated within the area. The genotype replacement event took place over five years. However, the underlying cause of the shift in the genotype, whether it was due to the viral fitness of DENV-3III or the human immune response, remains to be elucidated. In the case of the clade shift involving DENV-3III in Sri Lanka, viruses from the 1989 to 1998 period, which were associated with the country’s emergence of DHF, displaced the earlier viruses collected from 1981 to 1989 that were linked to milder cases [77]. The new virus demonstrated greater efficiency in infection and dissemination in A. aegypti [154]. Interestingly, this DENV-3III virus spread to the Americas in the 1990s [77]. The genotype replacement events subsequently led to the extinction of the genotypes that were previously in circulation, including DENV-2III (Asian-American), DENV-2IV (Asian-II), and DENV-3II.

Overall, DENV-4 has circulated less frequently compared to the other three serotypes but has persisted within the region, occasionally causing outbreaks. Interestingly, during periods of relatively low dengue incidence, DENV-4 has remained present and has tended to predominate. Since 2015, localized strains have reemerged, including DENV-4I in Vietnam [155], Laos [156], and India [157] and DENV-4IIA in Malaysia [158] and Indonesia [159]. These outbreaks have primarily occurred in specific areas rather than at a national scale—for example, the outbreaks that have occurred in cities in southern India, central Vietnam, and in East Java in Indonesia. These recurrent outbreaks of DENV-4 are largely attributable to a lack of immunity within the affected populations. Furthermore, most cases from 2018 in Vietnam and from 2020 to 2022 in Malaysia have primarily been associated with warning signs, while cases in India and Indonesia have generally been mild.

Since homotypic infection provides lifelong protection against the exposed serotype and temporary protection against other serotypes, it plays a role in DENV transmission and patient outcomes [160]. Currently, the co-circulation of the multiple of DENV types continues in many regions, which can promote antigenic variation and lead to the production of different antibodies. During secondary infection, sublevel heterotypic antibodies can influence severe dengue through antibody-dependent enhancement (ADE) [161,162]. Serological cohort studies were conducted in Thailand, Cuba, Vietnam, and Nicaragua with the aim of observing ADE [161,163,164,165]. The mechanism of ADE facilitates the entry of the virus through cross-reactive antibodies that interact with Fc gamma receptors expressed on target cells, including monocytes, macrophages, dendritic cells, and B cells [166], converting them to antigen-presenting cells. This interaction facilitates DENV infection and replication as well as cytokine production during ADE, potentially increasing vascular permeability and resulting in severe dengue [167,168,169]. On the other hand, a significant number of severe dengue cases were observed in primary infections, indicating that factors other than ADE are involved [170,171]. In severe cases, high levels of antibodies to the NS1 protein were detected in the critical phase [172]. The NS1 protein plays a role in severe pathogenesis by inducing vascular leakage, damaging epithelial cells, and triggering cytokine production [173], and its presence (NS1 antigenemia) is a critical marker for monitoring disease progression [174,175]. Moreover, periodic changes in serotype influence community immunity dynamics. A population that is naïve to the virus can experience serotype, genotype, or even clade replacements, leading to a resurgence of dengue cases [176,177]. Physical factors, including mosquito population size and density, are also important considerations. For example, in Singapore, despite efforts to control the vector, the long-term decline in DENV transmission has resulted in insufficient herd immunity, rendering populations more susceptible to epidemics [176,178]. Furthermore, antigenic changes among and within serotypes observed in Thailand over 20 years have resulted in a reduction in neutralization [4]. Genotype replacement events can also lead to a resurgence in dengue cases. Understanding the oscillation of DENV is important for a variety of reasons, including the development of vaccine programs, dengue epidemic prediction, and viral antigenic evolution.

## 6. Recent Emergence of Asian DENV Lineages into New Territories

Like the chikungunya virus (CHIKV), an arbovirus that has experienced multiple dispersals across various regions, DENV has exhibited similar dynamics. In 2004, CHIKV from coastal Kenya spread to adjacent islands, establishing a novel lineage that subsequently spread throughout the Indian Ocean, the Indian subcontinent, and Southeast Asia [179,180]. It is likely that this lineage was reintroduced to East Africa in recent years [181], as reported by us previously. Additionally, in 2013, the Asian CHIKV spread from Pacific islands to the Americas, resulting in rapid outbreaks in these regions [182,183].

Similarly, DENV-1III (or DENV-1V according to the Goncalvez [42] and Chen and Vasilakis classification [34]) was the predominant circulating type during the 2019 epidemic in Tanzania. The virus belonged to the Asian lineage, possibly originating from Singapore or India [184]. DENV-1III was also highly prevalent on Reunion Island from 2019 to 2021 [185]. Meanwhile, DENV-2II Cosmopolitan identified during 2018–2019 in Angola was from the Indian lineage rather than the African lineage that was circulating in West Africa [186]. The sequence indicated a close relationship to the virus present in the Democratic Republic of Congo in 2016 [187]. It is likely that the virus was introduced from the Indian subcontinent sometime earlier, leading to subsequent local transmission in this area [188].

The Americas are significantly affected by endemic DENV, with all four serotypes circulating across more than 40 countries. DENV invasions from Asia have included the DENV-2 Asian-American genotype, which was introduced from Southeast Asia in the 1980s and subsequently replaced the native American genotype [189]. Experimental studies on genotype fitness have shown that the DENV-2I American virus is less competitive than the DENV-2III Asian-American virus. The DENV-2I virus has a reduced capacity to infect and disseminate within A. aegypti populations [190]. In in vitro assays involving human cells, this virus replicates less efficiently in dendritic cells, which are important targets of virus replication [191]. Consequently, the American genotype was eventually displaced in the Americas region. Additionally, the DENV-3III genotype spread from Sri Lanka to Nicaragua and Panama in 1994, extended to the Caribbean, and then reached South America by 2000 [192]. These DENV strains established local lineages and have continued to be transmitted to the present. However, until recently, no other DENV invasions had been recorded from Asia for decades.

In recent years, there has been a noticeable increase in the introduction of various DENV-2II Cosmopolitan lineages into the Americas and the Caribbean, demonstrating their invasive potential. From 2019 to 2020, DENV-2II Cosmopolitan, which spread to Martinique in the Caribbean via France and India, was identified as belonging to the Indian lineage [193]. A strain of DENV-2II Cosmopolitan detected in Peru in 2019 was closely related to strains from Bangladesh in 2017–2018 [194] and was part of the Asian-Pacific lineage. This strain subsequently emerged widely across South America, including Brazil in 2021 as well as Colombia and Paraguay, and also emerged in Florida in the USA [195,196]. Furthermore, another distinct route was identified in which viruses from the Asian-Pacific lineage, commonly detected in India, China, and Thailand, spread to the French Caribbean Islands in 2023–2024, eventually reaching France in Europe and Florida, USA, by mid-2023 [197].

Several countries in Mainland SEA experienced the invasion of the DENV-2II Cosmopolitan Asian-Pacific lineage alongside the local genotype, DENV-2V Asian-I. This led to the simultaneous circulation of multiple genotypes during various epidemics or seasons thereafter, although complete replacement has not yet been observed. Similarly, the emergence of DENV-2II in areas previously endemic to DENV-2III Asian-American has resulted in ongoing co-circulation [198], and whether future replacement will occur remains uncertain. However, it is likely that pre-existing immunity developed against DENV-2III during the 2019 outbreak provided homotypic protection, particularly a few years after exposure.

DENV-3III, the active lineage circulating in the Indian subcontinent (India and Bhutan), was introduced to the Americas, with at least two introductions from Asia through the Caribbean: first in 2020 in Guadeloupe, Martinique, and Jamaica, and again in 2022 in Cuba. It was subsequently detected on other islands, Florida, and Brazil [193,199,200,201]. DENV hit the Americas with the highest ever recorded case numbers of 2.8 million and 4.6 million cases in 2022 and 2023, respectively, and 6 million as of May of 2024 [202]. Multiple serotypes were involved, but in several countries, DENV-3 was the predominant type, reappearing intermittently after several years [203]. For example, in Brazil, a DENV-3 epidemic was last reported in 2008 and it was rarely detected until 2022 [204].

The recent DENV epidemics in the Americas may be driven by several factors, including serotype turnover, the introduction of novel lineages, and genotype replacement events that have led to a lack of community immunity. Furthermore, rising temperatures and El Niño events have impacted both dengue and mosquito populations, extending the transmission season and expanding the transmission area into temperate regions [205,206,207,208]. The increased mobility and travel following the COVID-19 pandemic era have also facilitated the widespread dissemination of DENV across regions and continents [199]. For example, in 2022, the DENV-3III Asian lineage was identified on African islands in the Gulf of Guinea, with the index patient returning from the Caribbean [209]. There have also been DENV transmissions within France and French overseas territories in the Caribbean, the Pacific Ocean, and the Indian Ocean [193,197].

## 7. Conclusions

Dengue represents a significant global health and economic challenge. The frequency, magnitude, and geographical spread of dengue epidemics is increasing. Asian countries are hyperendemic, with all four serotypes and multiple genotypes in circulation. Recent changes in DENV epidemiology within this region have been driven by globalization, climate change, and virus evolution. This review analyzed public DENV sequence collections to clarify the distribution and phylogenetic relationships of DENV in Asian countries from the 1950s to 2023. The co-circulating serotypes have varied over time and differ by country. South Asia and Southeast Asia, including both mainland and maritime regions, have historically harbored distinct genotypes/lineages for extended periods. However, events such as viral introduction, replacement, and extinction have occurred; for example, the displacement of DENV-3II by DENV-3III led to the extinction of DENV-3II in the Mainland SEA region. The co-circulation of multiple genotypes continues in many countries. The spread of the DENV-2II Cosmopolitan genotype to Mainland SEA countries has established localized circulation, while the detection of the DENV-2II Cosmopolitan genotype of Asian-Pacific lineage in countries of the Indian subcontinent suggests the mixing of viruses from different geographic regions within the lineage. Despite limitations in the availability of DENV sequences for some countries and periods, which may affect the accuracy of interpretations, this study highlights the trends and patterns in DENV epidemiology. Our findings contribute to a better understanding of the complex dynamics of DENV in Asia.

## Figures and Tables

**Figure 1 viruses-16-01046-f001:**
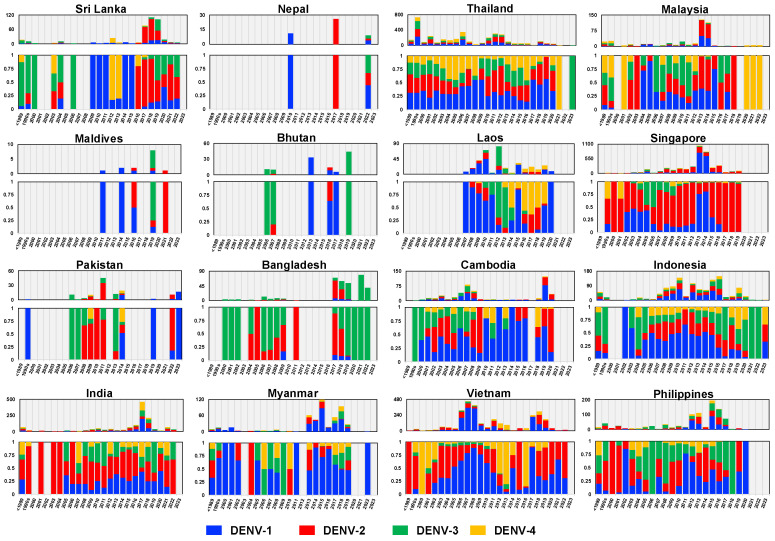
Circulation of DENVs in 16 Asian countries from the 1950s to 2023. DENV envelope sequences, sorted by serotypes and year, were collected from GenBank and ordered according to their country of origin. DENV cases were aggregated for sequences collected prior to 1989 and for those collected between 1990 and 1999. Top panel: Number of DENV cases per year by serotype. Bottom panel: Relative proportions of DENV cases per year by serotype.

## Data Availability

Data are contained within the review or Appendix A.

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
