# Peer review of "Unraveling Dengue Virus Diversity in Asia: An Epidemiological Study through Genetic Sequences and Phylogenetic Analysis"

_viruses, 2024, doi:10.3390/v16071046_

Round 1

Reviewer 1 Report

Comments and Suggestions for Authors

In this review manuscript, Phadungsombat, et al. analyzed DENV sequences in the public database to reveal the distribution and phylogenetic relationships of DENV in Asian countries from the 1950s to 2023. I congratulate the authors for preparing this important review. The manuscript is well-written and easy to be followed.

The study of dengue virus (DENV) genome dynamic is indeed important to understand better the evolutionary aspects of this virus in relation to its transmission, which can aid the better understanding of dengue disease.

The authors described the nomenclature for genotyping and mentioned the consistency of the currently used nomenclature with previously published classifications (Line 144-145). I wonder if the authors would argue to propose an updated classification for all DENV serotypes. Would the data from Asia be sufficient enough to propose the new classifications?

Striking serotype proportion for DENV-4 could be seen in almost all countries (and regions) studied. This serotype has always had the lowest proportion (except in Malaysia in 2020-2022 and Vietnam for some periods). It will be interesting to elaborate on this further and discuss more on what happened when DENV-4 predominates.  

The idea of the different genotype fitness has been mentioned (Line 787-788). It is arguably important to further elaborate on this using examples from previously published studies. Although still controversial in nature, I think this is an interesting point of discussion.

How about antibody-dependent enhancement (ADE) mechanisms (mentioned in line 796-802)? I feel that the authors only describe a little bit about this. It is important to correlate (if there’s any) between herd immunity, serotype/genotype replacement, and ADE/severity using examples from previous studies.

Minor comments:

1.      Abstract: dengue fever is one of the dengue manifestations (WHO 1997 classification). Here the term “dengue” as a disease is more appropriate than “dengue fever”. Additionally, it would also be useful to describe the other dengue 1997 and 2011 (https://iris.who.int/handle/10665/204894)  WHO classifications and to compare between these classifications (for example, a good reference such as Hadinegoro, SRS. Paediatr Int Child Health. 2012 May; 32(s1): 33–38. doi: 10.1179/2046904712Z.00000000052)

2.      The size of the text in x- and y-axis of Figure 1 is hard to read. Would there be a way to improve the readability of this figure?

3.      Line 134: “occurrence of recombination events that may lead to erroneous genotyping”. Please elaborate more on this. Does recombination event commonly occur (like in Influenza) and what is the rate of recombination event in dengue virus?

4.      Would it be confident enough to say that DENV-1I is replacing DENV1-IV in Southeast asia (SEA)?

5.      What happened to DENV1-II genotype? Was the disappearance of this genotype was caused by the lack of fitness?

6.      Genotype replacement in DENV-1 has been reported, i.e. genotype I replacing IV in Indonesia. Would this add to the “Temporal dynamics of DENV in South Asia and Southeast Asia” section of the manuscript?

7.      DENV genotypes differ between three geographical regions, especially between Mainland SEA in all four DENV serotypes. Could any geographical barriers cause this phenomenon? However, we know that extensive travel can easily overcome the geographical barriers.

8.      Line 792-802: How about the postulate of “herd/community immunity” and its role in genotype replacement events that can lead to a resurgence in dengue cases? The authors mentioned this in line 860.

Author Response

Response to Reviewer 1 Comments

Open Review

Quality of English Language

( ) I am not qualified to assess the quality of English in this paper
( ) English very difficult to understand/incomprehensible
( ) Extensive editing of English language required
( ) Moderate editing of English language required
( ) Minor editing of English language required
(x) English language fine. No issues detected

Is the work a significant contribution to the field?

Is the work well organized and comprehensively described?       

Is the work scientifically sound and not misleading?       

Are there appropriate and adequate references to related and previous work?     

Is the English used correct and readable?         

Comments and Suggestions for Authors

In this review manuscript, Phadungsombat, et al. analyzed DENV sequences in the public database to reveal the distribution and phylogenetic relationships of DENV in Asian countries from the 1950s to 2023. I congratulate the authors for preparing this important review. The manuscript is well-written and easy to be followed.

The study of dengue virus (DENV) genome dynamic is indeed important to understand better the evolutionary aspects of this virus in relation to its transmission, which can aid the better understanding of dengue disease.

We greatly appreciated comments and suggestions. Our dispositions to the reviewer’s comments are as follows;

The authors described the nomenclature for genotyping and mentioned the consistency of the currently used nomenclature with previously published classifications (Line 144-145). I wonder if the authors would argue to propose an updated classification for all DENV serotypes. Would the data from Asia be sufficient enough to propose the new classifications?

Response: We appreciate the Reviewer’s kind suggestion. While it is tempting to propose new classification of Indian and Asian-Pacific lineages of DENV2 genotype II, we are concerned that doing so would cause confusion in the field. Consequently, we have not attempted to propose a new classification in this study.

Striking serotype proportion for DENV-4 could be seen in almost all countries (and regions) studied. This serotype has always had the lowest proportion (except in Malaysia in 2020-2022 and Vietnam for some periods). It will be interesting to elaborate on this further and discuss more on what happened when DENV-4 predominates.  

Response: Thank you for this suggestion. As recommended, we have added additional text on DENV-4 dynamics (Lines 846-857).

The idea of the different genotype fitness has been mentioned (Line 787-788). It is arguably important to further elaborate on this using examples from previously published studies. Although still controversial in nature, I think this is an interesting point of discussion.

Response: Thank you for this suggestion. As recommended, we have supplemented the text with the findings of previous studies related to virus fitness in DENV-3III (Lines 838-843).

How about antibody-dependent enhancement (ADE) mechanisms (mentioned in line 796-802)? I feel that the authors only describe a little bit about this. It is important to correlate (if there’s any) between herd immunity, serotype/genotype replacement, and ADE/severity using examples from previous studies.

Response: Thank you for this suggestion. As recommended, we have provided a more description of  ADE in Lines 862-872.

Minor comments:

  1. Abstract: dengue fever is one of the dengue manifestations (WHO 1997 classification). Here the term “dengue” as a disease is more appropriate than “dengue fever”. Additionally, it would also be useful to describe the other dengue 1997 and 2011 (https://iris.who.int/handle/10665/204894)  WHO classifications and to compare between these classifications (for example, a good reference such as Hadinegoro, SRS. Paediatr Int Child Health. 2012 May; 32(s1): 33–38. doi: 10.1179/2046904712Z.00000000052)

Response: Thank you for this suggestion. As recommended, we have modified the text in Lines 8, 39-46 in the revised manuscript.

  1. The size of the text in x- and y-axis of Figure 1 is hard to read. Would there be a way to improve the readability of this figure?

Response:  As recommended, we have modified Figure 1 by increasing the font size and adjusting the scale.

  1. Line 134: “occurrence of recombination events that may lead to erroneous genotyping”. Please elaborate more on this. Does recombination event commonly occur (like in Influenza) and what is the rate of recombination event in dengue virus?

Response: We have provided more detail regarding the recombination events (Lines 148-151). However, as it is quite rare in DENV, and we could not provide the rate of recombination.

  1. Would it be confident enough to say that DENV-1I is replacing DENV1-IV in Southeast asia (SEA)?
  2. What happened to DENV1-II genotype? Was the disappearance of this genotype was caused by the lack of fitness?
  3. Genotype replacement in DENV-1 has been reported, i.e. genotype I replacing IV in Indonesia. Would this add to the “Temporal dynamics of DENV in South Asia and Southeast Asia” section of the manuscript?

Response: We appreciated comments 4, 5, and 6 from the Reviewer. Regarding DENV-1 dynamics, we added some text to the section “Temporal dynamics of DENV in South Asia and Southeast Asia” (Lines 798-809). Briefly, DENV-1I circulated across Southeast Asia, except in the Philippines and displacement has occurred Indonesia. The reason for the disappearance of DENV-1II is unclear; it may be due to a decrease in virus fitness or because it was collected from a niche environment.                    

  1. DENV genotypes differ between three geographical regions, especially between Mainland SEA in all four DENV serotypes. Could any geographical barriers cause this phenomenon? However, we know that extensive travel can easily overcome the geographical barriers.

Response: To our knowledge, DENV genotypes have historically exhibited distinct in geographical distributions influenced by various factors. However, this situation has changed in recent years. Important factors such as urbanization and increased population densities have facilitated the virus’ spread to adjacent areas and countries. Mosquito vectors are now capable of transmitting the virus across broader geographic regions. Furthermore, extensive travel contributes to the virus’ transmission to distant regions where populations are susceptible. Additionally, there exists a potential for the virus to infect susceptible individuals who lack pre-existing immunity.

  1. Line 792-802: How about the postulate of “herd/community immunity” and its role in genotype replacement events that can lead to a resurgence in dengue cases? The authors mentioned this in line 860.

Response: Thank you for this suggestion. As recommended, we have added this point in Lines 876-882.

Submission Date

31 May 2024

Date of this review

10 Jun 2024 22:25:19

Reviewer 2 Report

Comments and Suggestions for Authors

There are a couple of major issues with this study.

1) This manuscript is said to be a review but it seems to be based on an analysis done by the authors. It is unclear how this new analysis is different from or superior to the previous analyses. 

2) The phylogenetic trees shown in this work have no descriptions of the parameters used to create the trees. There are no support values for the nodes which make it very difficult to evaluate the authors' conclusions. It is impossible to determine the internal branching within terminal clusters because of the overlap of the terminal nodes.

3) The basics of how Genome Detective assigns genotypes and lineages needs to be included in order to evaluate the conclusions drawn.

4) The authors speak of different geographic regions but there is no description of what makes each region distinct or why the countries in the regions should be grouped together. Most of the descriptions are at the country level which makes the use of geographic regions unclear.

Comments on the Quality of English Language

The language will require some editing.

Author Response

Response to Reviewer 2 Comments

Open Review

Quality of English Language

( ) I am not qualified to assess the quality of English in this paper
( ) English very difficult to understand/incomprehensible
( ) Extensive editing of English language required
( ) Moderate editing of English language required
(x) Minor editing of English language required
( ) English language fine. No issues detected

Comments and Suggestions for Authors

There are a couple of major issues with this study.

We greatly appreciated comments and suggestions. Our dispositions to the reviewer’s comments are as follows;

1) This manuscript is said to be a review but it seems to be based on an analysis done by the authors. It is unclear how this new analysis is different from or superior to the previous analyses. 

Response: In the present study, we attempted to describe both historical and recent trends and patterns in DENV epidemiology in Asia using sequences from the very beginning of dengue virus identification until the most recent submissions to databases. In addition, we provided this information and compared it to previous DENV classifications. To the best of our knowledge, there is no precedent of such a comprehensive analysis. To emphasize this point, we slightly modified the abstract in the revised manuscript.

2) The phylogenetic trees shown in this work have no descriptions of the parameters used to create the trees. There are no support values for the nodes which make it very difficult to evaluate the authors' conclusions. It is impossible to determine the internal branching within terminal clusters because of the overlap of the terminal nodes.

Response: We apologize for this oversight. We have described the methodologies and parameters (Lines 158-162, and in the caption of Figures 3, 5, 7, and 9), as well as the support values for the main Figures 3, 5, 7, and 9. In addition, the support for internal branches have been inserted in Supplementary Figure S1A-S1D for DENV-1, Figure S2A-S2G for DENV-2, Figure S3A-S3D for DENV-3, and Figure S4A-S4D for DENV-4.

3) The basics of how Genome Detective assigns genotypes and lineages needs to be included in order to evaluate the conclusions drawn.

Response: Thank you for this suggestion. As recommended, we have added details to address this point (Lines 165-171).

4) The authors speak of different geographic regions but there is no description of what makes each region distinct or why the countries in the regions should be grouped together. Most of the descriptions are at the country level which makes the use of geographic regions unclear.

Response: Thank you for this suggestion. As recommended, we have clarified this issue in the revised manuscript (Lines 108-116).

Comments on the Quality of English Language

The language will require some editing.

Response:  We have had our revised manuscript proofread by a commercial English proof reader.

Submission Date

31 May 2024

Date of this review

13 Jun 2024 19:38:20
